# Dissecting the Functional Interplay Between Heme Oxygenase LjHO1 and Leghemoglobins in *Lotus japonicus* Nodules

**DOI:** 10.3390/biology14101401

**Published:** 2025-10-13

**Authors:** Yu Zhou, Tao Tian, Jie Ji, Liting Tan, Kexin Peng, Zhuocheng Liu, Wenlong Zhao, Chuanzhi Wang, Fawang Liu, Xingtao Zhang

**Affiliations:** 1School of Biological and Food Engineering, Suzhou University, Suzhou 234000, China; zhouyu@ahszu.edu.cn (Y.Z.); 19971820673@163.com (L.T.); 19855969348@163.com (K.P.); lzcdzsyx@163.com (Z.L.); zhaowenlong_2020@163.com (W.Z.); jfwcz@163.com (C.W.); fawang10314@ahszu.edu.cn (F.L.); 2National Key Laboratory of Agricultural Microbiology, Hubei Hongshan Laboratory, Huazhong Agricultural University, Wuhan 430070, China; taotian@webmail.hzau.edu.cn (T.T.); jieji@webmail.hzau.edu.cn (J.J.)

**Keywords:** heme, heme oxygenase (HO), leghemoglobin (Lb), *Lotus japonicus*, root nodule, symbiotic nitrogen fixation (SNF)

## Abstract

Legume root nodules require tightly regulated heme metabolism to support symbiotic nitrogen fixation (SNF). Leghemoglobins (Lbs) act as the primary heme reservoir, while heme oxygenase LjHO1 mediates heme degradation. In this study, we generated single and combinatorial mutants of *Lbs* and *LjHO1* in *Lotus japonicus* using the CRISPR/Cas9 system. Our results show that Lbs are essential for heme synthesis and nitrogen fixation, whereas LjHO1 fine-tunes heme turnover and contributes to nodule formation. These findings highlight the complementary roles of Lbs and LjHO1 in maintaining nodule heme homeostasis and symbiotic efficiency.

## 1. Introduction

Legumes establish a specialized symbiosis with nitrogen-fixing rhizobia, resulting in the formation of nodules where atmospheric nitrogen is reduced to ammonia. This process supplies the host with an essential source of nitrogen while simultaneously enriching soil fertility, reducing the need for synthetic fertilizers and promoting sustainable agriculture [1,2,3]. A defining feature of functional nodules is the establishment of a microaerobic environment that simultaneously protects oxygen-labile nitrogenase from inactivation and provides sufficient oxygen for bacteroid respiration [4]. Leghemoglobins (Lbs), a class of nodule-specific hemoglobins, are indispensable for maintaining this oxygen balance [5]. By reversibly binding oxygen, Lbs facilitate its efficient delivery to bacteroids while buffering free oxygen concentrations to safeguard nitrogenase activity.

Lbs are extraordinarily abundant, comprising approximately 25–30% of the total soluble protein content in mature nodules [6]. Their prosthetic heme groups account for the vast majority of the nodule heme pool, linking nodule physiology tightly to heme metabolism. In *Lotus japonicus*, three *Lb* genes (*Lb1*, *Lb2*, and *Lb3*) exist in the genome [4], and simultaneous knockout of all three isoforms (the triple mutant, *lb123*) leads to severe defects in symbiotic nitrogen fixation (SNF), premature nodule senescence, and a rapid decline in heme content [4,7]. Beyond impairing oxygen transport, the loss of *Lbs* perturbs cellular homeostasis, including disrupted mitochondrial cristae and accumulation of reactive oxygen species (ROS) [7], likely resulting from mis-accumulated free heme and its cytotoxic effects. Recent studies further revealed that *Lb* expression is transcriptionally regulated by NIN (Nodule Inception) and NLP (NIN-like proteins), establishing a direct genetic link between nodule developmental regulators and oxygen-buffering capacity [8]. Moreover, *Lb* gene expression is highly dynamic, responding not only to nodule developmental stages but also to environmental cues such as nitrate availability and dark stress, thereby highlighting their pivotal role in coordinating symbiotic performance with external conditions [9].

Massive heme biosynthesis is required during nodule development to meet the demand for Lb assembly [10]. The tetrapyrrole pathway, which provides precursors for heme, is highly active in nodules [11,12,13,14]. In early nodule development, *L. japonicus* glutamyl-tRNA reductase (GluTR), a key rate-limiting enzyme in this pathway, is responsible for synthesizing approximately 75% of the heme in nodules, most of which is used for the assembly of cytoplasmic apo-Lbs [11,15]. In mature nodules, Lbs accumulate to extraordinarily high levels, representing the majority of the soluble heme pool. However, as nodules senesce, proteases may release heme from Lbs [16,17], and this free heme has been implicated in promoting lipid peroxidation and symbiosome membrane degradation [18]. Therefore, tight coordination of heme biosynthesis, incorporation into Lbs, and controlled release during nodule development and senescence is crucial for maintaining nodule integrity and symbiotic function.

The balance between heme biosynthesis, utilization, and degradation is critical for maintaining nodule function. Heme oxygenases (HOs) constitute the only known enzymatic pathway for heme catabolism in plants, catalyzing the oxidative cleavage of heme into biliverdin (BV), carbon monoxide and ferrous iron (Fe^2+^) [19,20,21]. In higher plants, heme oxygenase plays a role in the biosynthesis of the bilin chromophore for light perception, the mobilization of iron under deficiency, and antioxidant defence against cellular stress [22,23,24,25]. In *L. japonicus*, *heme oxygenase 1* (*LjHO1*) is the predominant nodule-expressed heme oxygenase, accumulating in mature nodules and further upregulated during senescence [10]. The nodule-enriched expression of *HO1* in *Medicago sativa* and the retention of HO1 activity in the nitrogen-fixing organelles (nitroplast) of marine algae indicate that the role of HO in heme catabolism is evolutionarily conserved [26,27]. Furthermore, recent studies revealed that a bZIP transcription factor FUN directly targets the *LjHO1* promoter to mediate nitrate-induced nodule senescence, linking HO function to zinc-dependent regulation of symbiosis [28]. The senescence-associated induction of HO1 suggests that it acts as a safeguard, preventing the toxic accumulation of toxic free heme released from degraded Lbs.

Lbs play a dual role as both the primary heme reservoir and dominant sink. However, the dynamic interactions among Lb abundance, *HO1* expression and heme flux during different stages of nodule development and senescence remain poorly understood. In this study, we show that Lbs loss strongly reduces heme levels, *LjHO1* deficiency slightly elevates heme in *Lb*-deficient nodules, and their combined disruption impairs nodule formation and nitrogenase activity. These findings reveal how legumes dynamically coordinate Lb abundance and *HO1* expression to sustain SNF while preventing heme toxicity.

## 2. Materials and Methods

### 2.1. Plant Material, Bacterial Strains and Growth Conditions

The MG20 ecotype of *Lotus japonicus* was used as WT in this study. The *ho1-1* [10] and *lb123-1* mutant [7] has been described previously. Briefly, *ho1-1* carries a loss-of-function mutation in *LjHO1* leading to reduced heme degradation, whereas *lb123-1* is a triple knockout mutant lacking *Lb1*, *Lb2*, and *Lb3*, resulting in severely impaired leghemoglobin accumulation and nitrogen fixation capacity. For seed germination, the seeds were immersed in concentrated sulfuric acid for 10 min, and then surface sterilized with 10% (w/v) sodium hypochlorite for 5 min. After being rinsed five times with sterile water, seeds were imbibed in water at 4 °C for 1 d. Then seeds were transferred to one-half strength Murashige-Skoog medium and grown in the dark for 2 d at 22 °C, followed by a 16 h light/8 h dark cycle for another 3 d. For the nodulation assay, the seedlings were planted in the pots containing sterile perlite–vermiculite (1:3) and supplemented with B&D medium containing 0.5 mM KNO_3_ [29] in a growth chamber at 22 °C under a long day photoperiod (16 h light; 120 μE intensity). After 5 days of growth, the seedlings were inoculated with either *Mesorhizobium loti* MAFF303099 or an mCherry-labelled derivative to assess symbiotic phenotypes [30,31].

### 2.2. Hairy Root Transformation in L. japonicus

Hairy root transformation was performed as previously described [32] with some modifications as described below. In brief, 5-day-old MG20 seedlings were cut at the point where the pigment in the hypocotyl begins to deposit, and then the seedlings were placed in *Agrobacterium rhizogenes* suspension containing 10 μg/mL acetosyringone. After being incubated at room temperature for 30 min, the seedlings were transferred onto 1/2 MS containing 10 μg/mL acetosyringone and kept their hypocotyl touching the medium. After co-cultivating for 5 days in a growth cabinet (24 °C dark for the first 2 days, 24 °C 16 h light/8 h dark cycle for next 3 days), the plants were transferred onto HRE agar medium containing 300 μg/mL Timentin and placed vertically in a growth cabinet (24 °C 16 h light/8 h dark cycle) and grown for 12 d. The seeding with positive transgenic hairy roots was planted in the pots containing sterile perlite–vermiculite (1:3) and supplemented with B&D medium containing 0.5 mM KNO_3_ in a greenhouse (24 °C, 16 h light/8 h dark cycle). After 5 days of growth, the plants were inoculated with *M. loti* MAFF303099 or a strain constitutively expressing mCherry.

### 2.3. GUS Staining

The construction of vectors for monitoring *LjHO1* promoter activity was performed as previously described [10]. The 2.9 kb promoter region of *LjHO1* was cloned into DX2181G-mCherry vectors harboring the *GUS* gene, and the resulting construct was introduced into *L. japonicus* roots via Agrobacterium-mediated hairy root transformation for GUS staining analysis. For GUS staining, *L. japonicus* nodules were collected at 2, 4, and 6 weeks post-inoculation with rhizobia, thoroughly washed to remove residual growth medium, and blotted dry. Samples were fixed in 90% (v/v) acetone for 20 min on ice at 4 °C, rinsed twice with 0.1 M phosphate-buffered saline (PBS, 5 min each), and subsequently incubated in GUS staining solution. Nodules were vacuum-infiltrated at −12 psi and stained at 37 °C in the dark, with incubation time adjusted according to nodule size. Specifically, nodules of different sizes were stained together with their associated roots: large nodules (~1 mm in diameter) were incubated for 5 h, while medium-sized nodules (~0.6 mm in diameter) were incubated for 3 h. A total of eight independent root transformations were analyzed for reproducibility. After staining, tissues were washed four times with 70% ethanol. For early infection stages, samples were cleared with a chloral hydrate–glycerol–water (8:2:1, w/v/v) solution before observation, whereas mature nodules were embedded in agarose, sectioned, and examined microscopically to visualize internal GUS activity.

### 2.4. Subcellular Localizations

For subcellular localization in *L. japonicus* nodules, the full-length *LjHO1* coding sequence without the stop codon was amplified from cDNA by PCR using specific primers (Appendix A). The amplification products containing *BamH*I and *Kpn*I sites were fused in frame with the coding region of sGFP harbored in a pUB vector to generate the recombinant plasmids. The *LjUBIQ* promoter was used to drive the expression of the fusion protein. The constructed vectors were transferred to MG20 via *R. rhizogenes* transformation. The transgenic plants were inoculated with *M. loti* MAFF303099 constitutively expressing mCherry and watered with B&D medium containing 0.5 mM KNO_3_. At 4 weeks post inoculation, nodules were sectioned and visualized under a confocal microscope Leica SP8 TCS (Leica, Wetzlar, Germany). GFP fluorescence was visualized using a 488 nm excitation wavelength, and emission was collected between 500 and 550 nm, while the mCherry fluorescence from rhizobia was detected using 588 nm excitation and 610–630 nm emission settings.

### 2.5. CRISPR-Cas9-Mediated Genome Editing in L. japonicus

Target-specific guide RNAs (gRNAs) were designed using online CRISPR design tools (CRISPR-P, http://cbi.hzau.edu.cn/CRISPR2/ (accessed on 15 September 2025)) to minimize potential off-target effects. Primer sequences used for gRNAs are listed in Appendix A. The gRNA sequences were cloned into a CRISPR-Cas9 binary vector under the control of the U6 promoter, while Cas9 was expressed under the CaMV *35S* promoter as described before [33]. The recombinant constructs were introduced into Agrobacterium tumefaciens for stable transformation of *L. japonicus* explants following established protocols [34]. Transformed plants were selected using appropriate selectable markers, and genomic DNA was extracted from leaves to verify mutations at the target loci via PCR and sequencing. Edited plants were further propagated to obtain T_2_ lines for downstream phenotypic and molecular analyses.

### 2.6. Acetylene Reduction Assay

Briefly, five to six nodulated roots detached from *L. japonicus* plants were collected and placed in a 30 mL glass tube with a rubber cap. Acetylene (2 mL) was injected into the tube in which water-absorbing cotton was placed on the bottom. After incubating at 28 °C for 2 h, the amount of ethylene produced was measured using a gas chromatograph (East and West Analytical Instruments, Inc., Beijing, China).

### 2.7. Heme Extraction and UPLC-MS/MS Analysis

A Thermo Scientific™ Ultimate™ 3000 LC system coupled to a Thermo Scientific™ TSQ Quantiva™ Triple Quadrupole mass spectrometer (Thermo Fisher Scientific, Milan, Italy) was used for the determination of heme concentrations. 0.03 g of frozen nodules material was ground up using a precooled pestle using an electric rotary pestle (TIANGEN, Beijing, China), and the obtained pellet was suspended in 0.2 mL of acetonitrile–acetic acid, 4:1 (v/v). After centrifuging at 10,000× *g* for 10 min at 4 °C, the suspension was transferred to a new 1.5 mL centrifuge tube and kept in the dark at 4 °C until used for UPLC/ESI-MS/MS. For the determination of the target compounds, 5 μL of the supernatant was injected into the reverse phase UPLC C18 column (Phenomenex Ultracarb (Phenomenex, Torrance, CA, USA); 4.6 × 250 mm; 5 μm ODS) with a 4.6 mm × 30 mm guard column of the same material and separation was carried out at a flow rate of 1 mL/min with 0.1% (v/v) formic acid aqueous solution as mobile phase A and 100% acetonitrile as mobile phase B. For the elution program, the following proportions of solvent B were used: 0–2 min, 10–30% B; 2–53 min, 30–98% B; 53–55 min, 98% B; 55–57 min, 98–10% B; 57–60 min, 10% B. Peak detection was at 405 nm. The mass spectrometer parameters were as follows: spray voltage: 3.5 kV (positive ion mode); 2.7 kV (negative ion mode); vaporizer temperature: 250 °C; sheath gas pressure: 35 arb. units; auxiliary gas flow: 10 arb. units; ion sweep gas pressure: 0 arb. units; capillary temperature, 350 °C; vaporizer temperature: 400 °C; The argon gas collision-induced dissociation was used with a pressure of 1.5 mTorr. The mass acquisition was performed in negative and positive ionization mode over a range of 100–1000 *m*/*z*.

### 2.8. Statistical Analysis

Statistical analyses were performed using one-way ANOVA followed by appropriate multiple comparison tests. Differences were considered statistically significant at *p* < 0.05.

## 3. Results

### 3.1. LjHO1 Expression Dynamics Under Lb Deficiency

Previous studies have shown that the transcript level of *LjHO1* is markedly elevated in *Lb*-deficient mutants [7]. To validate this regulatory feature, we generated a *pLjHO1:LjHO1-GUS* reporter construct and introduced it into wild-type (WT) MG20 and the *lb123-1* mutant via hairy root transformation. At 2 wpi, GUS staining was barely detectable in WT but strongly induced in *lb123-1* (Figure 1A,D), reflecting enhanced promoter activity in the *Lb*-deficient mutant background at the early stage. As development proceeded, *LjHO1* expression in WT nodules became progressively stronger (Figure 1B), while in *lb123-1* the staining intensity gradually faded (Figure 1E). When nodules entered the senescence phase, the LjHO1-GUS signal in WT reached its maximum level (Figure 1C), whereas only a faint residual expression could still be observed in *lb123-1* nodules (Figure 1F). In summary, *Lb* deficiency promotes *LjHO1* expression at early nodulation stages but fails to sustain its activity as nodules undergo premature senescence.

### 3.2. Lbs Deficiency Does Not Alter LjHO1 Localization in Uninfected Cells

Previous studies demonstrated that LjHO1 predominantly localizes to plastids of uninfected cells within nodules [10]. To further assess whether the complete loss of *Lb* genes (*Lb1*/*Lb2*/*Lb3*) affects their subcellular targeting, we expressed a *pLjHO1:LjHO1-sGFP* construct in transgenic hairy roots and inoculated them with *M. loti* stably expressing mCherry. Confocal microscopy revealed punctate LjHO1-sGFP signals that were clearly detected in plastids of uninfected cells in both WT MG20 and the *lb123-1* mutant nodules at 4 wpi (Figure 2A,B), confirming the cell-type specificity of LjHO1 localization. Collectively, these findings demonstrate that LjHO1 targeting and subcellular distribution remain unaltered in the absence of *Lbs*.

### 3.3. Phenotypic Analysis of the ho1lb123-1 Quadruple Mutant

To investigate the functional relationship between LjHO1 and Lbs during symbiotic nitrogen fixation (SNF) in *L. japonicus*, a quadruple mutant *ho1lb123-1* was generated in the *lb123-1* background using CRISPR/Cas9. Phenotypic analysis at 6 weeks post inoculation with *M. loti* MAFF303099 revealed that nodules of WT MG20 and *ho1-1* were typical pink, whereas those of *lb123-1* and *ho1lb123-1* were greenish (Figure 3A). Scoring of nodulation phenotypes showed that *ho1-1* exhibited a 15.7% reduction in nodule number and a 28.3% decrease in nitrogenase activity (Figure 3B,C), accompanied by pronounced reductions in shoot length and shoot fresh weight compared with the MG20 (Figure 3D,E). Both *lb123-1* and *ho1lb123-1* displayed pronounced decreases in nodule number, nitrogenase activity, shoot height and shoot fresh weight relative to MG20 (Figure 3B–E). Although most symbiotic traits did not differ significantly between *lb123-1* and *ho1lb123-1* (Figure 3C–E), the quadruple mutant exhibited a further 17.1% reduction in nodule number compared with *lb123-1* (Figure 3B), consistent with the phenotypic trend of the *ho1-1* mutant. Collectively, *Lb* deficiency largely underlies the impaired nitrogen fixation, while *LjHO1* deficiency further exacerbates the reduction in nodule formation capacity.

### 3.4. Loss of LjHO1 Slightly Elevates Heme Levels in ho1lb123 Mutant Nodules

Previous studies have established that LjHO1 primarily mediates heme degradation in nodules [10]. To further dissect the roles of LjHO1 and Lbs in maintaining heme homeostasis, we quantified heme levels in nodules of different genotypes using ultra-performance liquid chromatography coupled tandem mass spectrometry (UPLC-MS/MS). Chromatographic analysis revealed that heme from MG20 and the *ho1-1*, *lb123-1*, and *ho1lb123-1* mutants (6 wpi) eluted at 20.90 min, consistent with the standard (Figure 4A–E). Moreover, characteristic secondary MS/MS fragments (*m*/*z* 616.2→557.22; *m*/*z* 557.22→498.13) matched the standard exactly (Figure 4F–K), validating the efficiency of the acetonitrile–acetic acid (4:1, v/v) extraction protocol for tetrapyrrole molecules in nodules. Quantitative analysis showed a slight accumulation of heme in *ho1-1* nodules relative to MG20 (Figure 4L), indicating that *LjHO1* deficiency impairs heme degradation. By contrast, nodules of *lb123-1* and *ho1lb123-1* exhibited a pronounced reduction in heme levels compared with MG20 nodules (Figure 4L), indicating that loss of *Lbs* exerts a stronger inhibitory effect on heme accumulation. However, the heme content in *ho1lb123-1* was over 60% higher than in *lb123-1* (Figure 4L). Together, these results suggest that loss of *LjHO1* may lead to a tendency for heme accumulation in nodules lacking *Lbs*.

## 4. Discussion

Our study demonstrates that Lb exerts dynamic regulation over *LjHO1* expression during nodule development. In early-stage WT nodules, *LjHO1* expression is relatively weak due to the extensive sequestration of heme by Lb [10], which limits the pool of unbound heme (free heme). During nodule senescence, Lb degradation releases heme [10,35], thereby inducing *LjHO1* upregulation (Figure 1C). In the *lb123* mutant, *Lb* deficiency results in premature nodule senescence. Although the expression of heme biosynthetic genes is downregulated [7], a fraction of synthesized heme remains unbound, leading to an accumulation of free heme within nodule cells and a pronounced induction of *LjHO1* in early-stage *lb123* nodules (Figure 1E). These observations indicate that *LjHO1* expression is primarily determined by free heme availability, with Lb, as the major heme reservoir, controlling the pool of unbound heme in nodules.

In wild-type nodules, LjHO1 contains a plastid-targeting peptide, directing its localization to nodule plastids of uninfected cells (Figure 2A). This distribution aligns with the oxygen requirement for heme degradation and the need to spatially segregate the degradation product carbon monoxide from functional Lb in infected cells. To complement our findings, we explored publicly available transcriptome datasets. The transcript abundance of *MtHO1* (Medtr8g019320), the ortholog of *LjHO1* in *Medicago truncatula*, was retrieved from the MtGEA database and showed enhanced expression in young nodules (Appendix A). In addition, we analyzed a published single-nucleus transcriptome of soybean nodules [36] and identified two members of the soybean HO1 family (*GmHO1* and *GmHO3*). Both genes were mainly expressed in uninfected cell clusters (Appendix A). In *lb123* mutants, LjHO1 remains restricted to uninfected cells, indicating that its spatial patterning is independent of Lb gene function. Accordingly, at least in determinate nodules, heme synthesized in infected cells can still be transported to uninfected cells for degradation.

Phenotypic analyses revealed no significant differences between *lb123* and *ho1lb123* in nodule nitrogenase activity, shoot length, and shoot fresh weight (Figure 3C–E), suggesting that, in the absence of *Lb*, additional loss of *LjHO1* exerts minimal impact on overall nitrogen fixation and plant growth. This likely reflects that *Lb* deficiency already substantially compromises nodule function, rendering further loss of *LjHO1* largely inconsequential at the whole-plant level. However, the quadruple mutant showed a further reduction in nodule number compared with *lb123* (Figure 3B), consistent with trends observed in *ho1* mutants, indicating that LjHO1 retains a contributory role in nodule formation. Grafting experiments have shown that the reduced nodule number in *ho1* mutants is attributable to the loss of LjHO1 function in the shoot rather than the root [10]. Accordingly, LjHO1 may exert a systemic regulatory effect, potentially through shoot-derived signals that influence nodule initiation and development.

Previous studies have demonstrated that ROS can facilitate non-enzymatic heme degradation [37]. Consistently, NBT and DAB staining revealed abundant ROS, including superoxide radicals and hydrogen peroxide, in *lb123* nodules, and high levels of superoxide radicals in *ho1* nodules [7,10]. These observations raise the question of whether ROS contribute to heme degradation in these mutant nodules, and potentially even in naturally senescing nodules. In contrast to enzymatic degradation mediated by HO1, which specifically cleaves the α-methene bridge of the heme tetrapyrrole to produce biliverdin IXα (BV-IXα), ROS attack all carbon–methene bridges non-specifically, generating a variety of pyrrole products in addition to releasing iron. Therefore, monitoring the presence of biliverdin isomers other than BV-IXα could serve as an indicator of ROS-mediated heme degradation in nodules.

In this study, we generated quadruple *ho1lb123* mutants using CRISPR/Cas9 and incorporated them with previously established *ho1* and *lb123* mutants [7], providing robust experimental tools to systematically dissect the roles of Lb and LjHO1 in heme biosynthesis, degradation, transport, and regulation within nodules. These mutants not only elucidate the function of LjHO1 in maintaining heme homeostasis but also provide a framework for investigating the metabolic and signaling roles of Lb in the symbiotic nitrogen-fixing environment, offering critical insights into the molecular mechanisms underlying SNF. Recent progress in cell-type-specific CRISPR technology in legumes [38] provides a promising avenue for dissecting the precise contributions of genes like LjHO1 in specific nodule cell populations. Such targeted manipulations may enable fine-tuned engineering of heme metabolism, thereby offering new strategies to enhance SNF efficiency.

## 5. Conclusions

This study reveals a regulatory interplay between Lbs and HO1 in nodule heme homeostasis. Lbs act as the major heme reservoir, while LjHO1 fine-tunes degradation to prevent toxic heme accumulation. We show that *Lb* deficiency transiently induces *LjHO1* expression, whereas combined loss of both components disrupts heme balance, nodule development, and nitrogenase activity. LjHO1 localization to uninfected cells further suggests an intercellular pathway for heme turnover during senescence. These findings establish a dynamic balance between heme synthesis and degradation as a key determinant of effective SNF, advancing our understanding of heme metabolism and cofactor regulation in legume–rhizobium symbiosis.

## Figures and Tables

**Figure 1 biology-14-01401-f001:**
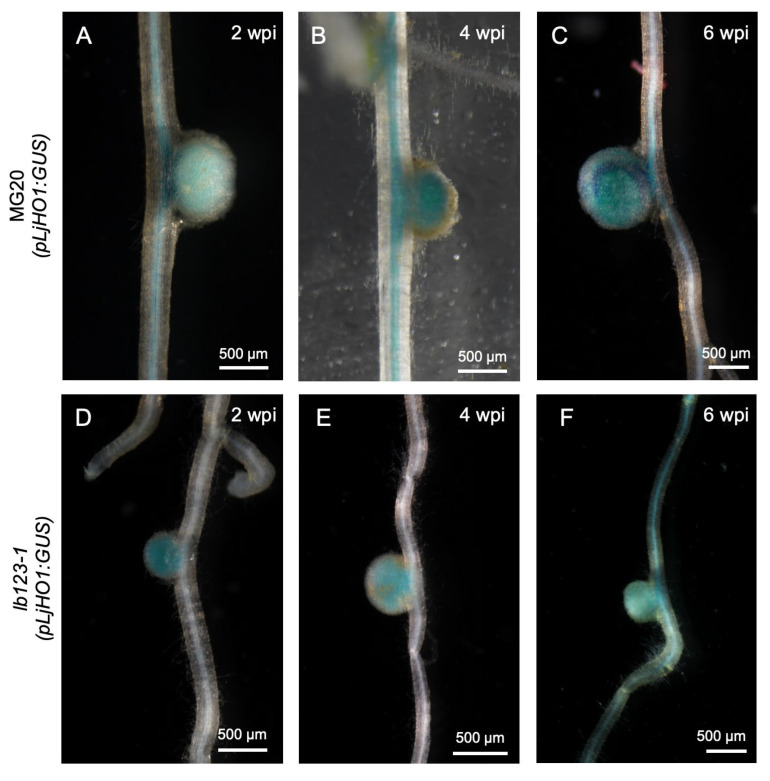
Spatiotemporal expression pattern of *LjHO1* in the *lb123-1* mutant background. Histochemical analysis of *pLjHO1:LjHO1-GUS* activity in nodules of WT MG20 (**A**–**C**) and the *lb123-1* mutant (**D**–**F**) at 2, 4, and 6 weeks post-inoculation (wpi) with *Mesorhizobium loti* MAFF303099. Representative images are shown from at least eight independent hairy root transformation events. Scale bar, 500 μm.

**Figure 2 biology-14-01401-f002:**
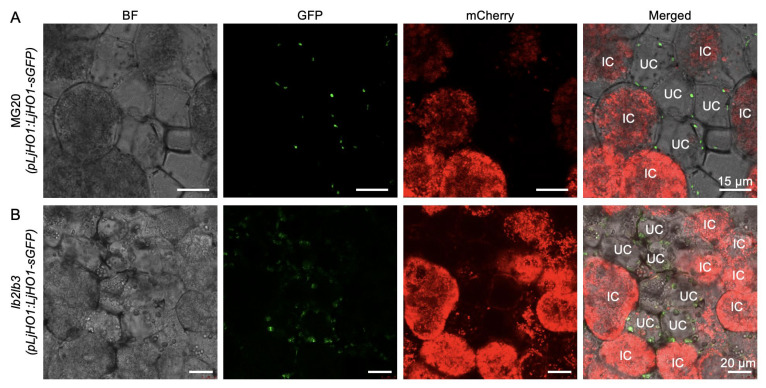
Subcellular localization of LjHO1 in MG20 and *lb123* mutant nodules. Transgenic hairy roots expressing *pLjHO1:LjHO1-sGFP* in MG20 (**A**) and *lb123-1* (**B**) were inoculated with *M. loti* MAFF303099 constitutively expressing mCherry. Nodule sections were examined at 4 weeks post-inoculation (wpi) by confocal microscopy. Merged images show GFP fluorescence, rhizobia (mCherry), and bright-field (BF) signals. Representative images are shown from at least six independent hairy root transformation events. IC, infected cell; UC, uninfected cell. Scale bars, 15 μm and 20 μm in (**A**) and (**B**), respectively.

**Figure 3 biology-14-01401-f003:**
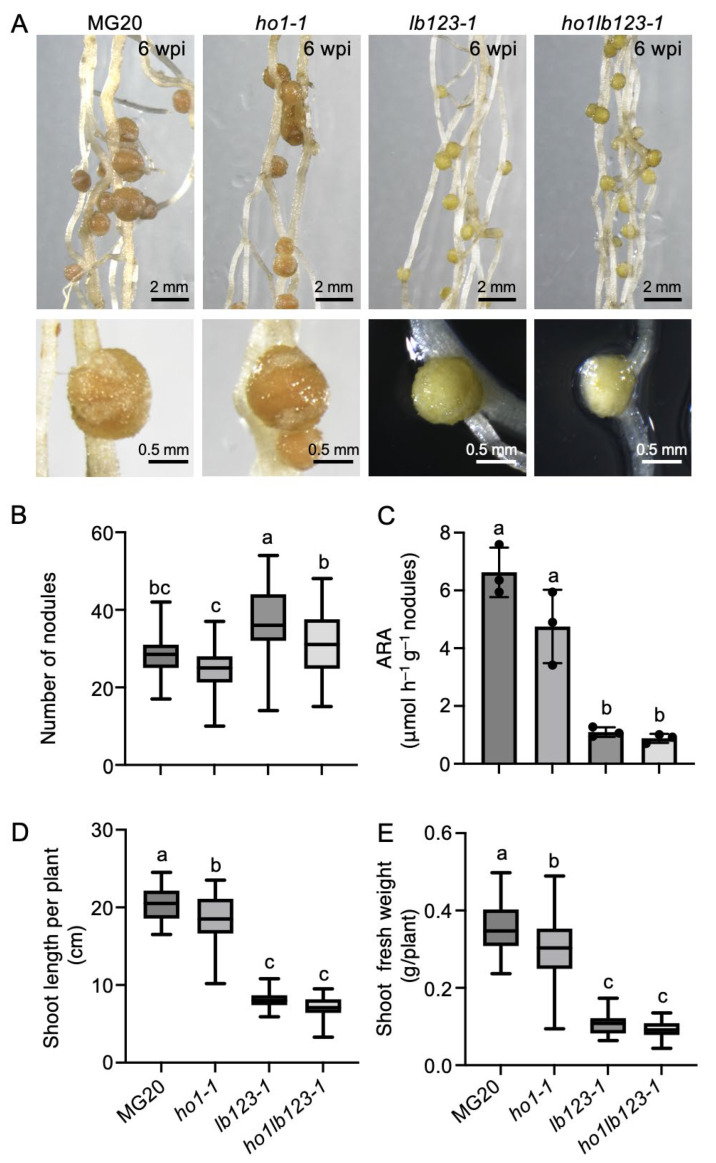
Symbiotic phenotypes of the *ho1lb123* quadruple mutant. (**A**) Representative nodules of MG20, *ho1-1*, *lb123-1*, and *ho1lb123-1* at 6 weeks post inoculation (wpi) with *M.loti* MAFF303099. Scale bar, 2 mm and 0.5 mm in the upper and lower panels, respectively. (**B**) Nodule number. Data are presented as mean ± SD (*n* ≥ 32). (**C**) Acetylene reduction activity (ARA). Data are shown as mean ± SD from three biological replicates, each consisting of nodules collected from 6 to 8 plants. (**D**) Shoot length. Data are presented as mean ± SD (*n* ≥ 30). (**E**) Shoot fresh weight. Data are presented as mean ± SD (*n* ≥ 30). For panels (**B**–**D**), statistical significance was determined using one-way ANOVA with multiple comparisons. Means labelled with the same letter are not significantly different at *p* < 0.05.

**Figure 4 biology-14-01401-f004:**
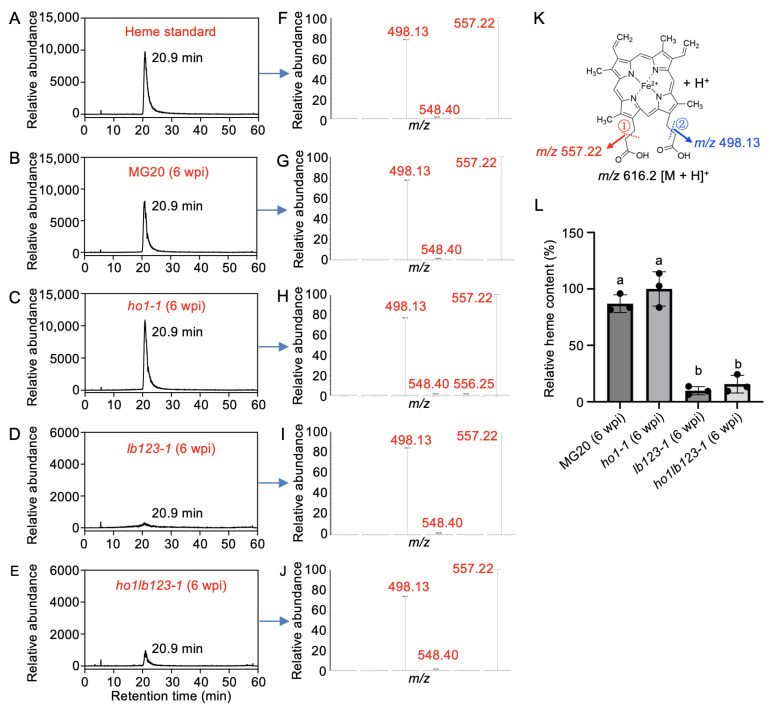
Quantification of heme content in the *ho1lb123* quadruple mutant. (**A**–**J**) UPLC chromatograms of the heme standard (**A**) and nodule extracts from MG20, *ho1-1*, *lb123-1*, and *ho1lb123-1* mutants at 6 wpi (**B**–**E**), along with the corresponding MS/MS spectra (**F**–**J**). The retention time of heme was 20.9 min. (**K**) Fragmentation pattern of heme showing the generation of ions at *m*/*z* 557.22 and *m*/*z* 498.13 in (**F**–**J**); ① and ② indicate the sequential cleavage sites of chemical bonds. (**L**) Quantitative analysis of heme levels in nodules from MG20, *ho1-1*, *lb123-1*, and *ho1lb123-1* at 6 wpi using UPLC/ESI-MS/MS. Data represent means ± SD of three biological replicates, each replicate consisting of nodules pooled from six plants. Statistical significance was determined using one-way ANOVA followed by multiple comparison tests. Means labelled with the same letter are not significantly different at *p* < 0.05.

## Data Availability

The data that support the findings of this study are available in the article and its Appendix A.

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
