# Peer review of "Dissecting the Functional Interplay Between Heme Oxygenase LjHO1 and Leghemoglobins in Lotus japonicus Nodules"

_biology, 2025, doi:10.3390/biology14101401_

Round 1
Reviewer 1 Report
Comments and Suggestions for Authors
The manuscript by Zhou et al. investigates the interaction between HO1 and leghemoglobins (Lbs) in Lotus japonicus nodules. Building on the previously characterized role of the Lb triple mutant, the authors examine the expression pattern of LjHO1 and generate a quadruple mutant. Through phenotypic and metabolic analyses, they propose that HO1 and Lbs play complementary roles in maintaining heme homeostasis within nodules. Overall, this study presents insights into nodule physiology and heme metabolism.
Below are some suggestions for improving the manuscript.
- Figure 1. Lots of published gene expression data during legume nodulation, including both bulk and single-cell RNA-seq datasets. The authors could leverage these resources to compile and compare the expression patterns of HO1 and Lb genes across legume species during nodulation. Presenting this information in a supplementary table or figure would usefully complement their experimental findings.
- A clear discrepancy exists between the weak pLjHO1-LjHO1-GFP signal in lb123 mutant in Figure 2 and the strong upregulation of pLjHO1-GUS expression reported in Figure 1. This contradiction needs to be discussed and clarified by the authors.
- The reference lack some recent key publications. For example, the study demonstrating that NIN and NIN-like proteins regulate Lb expression (PMID: 34709882). Research on the dynamics of hemoglobins during nodule development, nitrate response, and dark stress in Lotus japonicus (PMID: 37976184).
- The Discussion could be further strengthened. One major function of HO and Lb is regulating oxygen levels to support efficient nitrogen fixation. However, the authors did not discuss how oxygen availability may vary across the different mutants, nor its potential correlation with ARA nitrogenase activity. Furthermore, cell-type-specific CRISPR has been applied in legumes (PMID: 40641645), it would be valuable for the authors to discuss the potential of employing such advanced gene-editing technologies to achieve more precise manipulation of symbiotic nitrogen fixation efficiency in future studies.
Minor comments and edits:
Line13-14. This sentence is confusing and should be revised.
Line 89. reference is missing.
Line 122. The description of the GUS staining is unclear. It would be helpful to specify the incubation times for each nodule size category and the total number of roots analyzed.
Figure 2B. The lb2lb3 should be lb123-1.
Comments on the Quality of English LanguageThe manuscript contains many minor errors, including italics and spacing.
Author Response
|
The manuscript by Zhou et al. investigates the interaction between HO1 and leghemoglobins (Lbs) in Lotus japonicus nodules. Building on the previously characterized role of the Lb triple mutant, the authors examine the expression pattern of LjHO1 and generate a quadruple mutant. Through phenotypic and metabolic analyses, they propose that HO1 and Lbs play complementary roles in maintaining heme homeostasis within nodules. Overall, this study presents insights into nodule physiology and heme metabolism. Below are some suggestions for improving the manuscript.
Figure 1. Lots of published gene expression data during legume nodulation, including both bulk and single-cell RNA-seq datasets. The authors could leverage these resources to compile and compare the expression patterns of HO1 and Lb genes across legume species during nodulation. Presenting this information in a supplementary table or figure would usefully complement their experimental findings.
We appreciate the reviewer’s suggestion. To complement our findings, we explored publicly available transcriptome datasets. The transcript abundance of MtHO1 (Medtr8g019320), the ortholog of LjHO1 in Medicago truncatula, was retrieved from the MtGEA database and showed enhanced expression in mature nodules (Fig. X1A), consistent with the expression pattern of LjHO1 reported in our previous study (Zhou et al., 2023). In addition, we analyzed a published single-nucleus RNA-seq dataset of soybean nodules (Liu et al., 2023) and identified two members of the soybean HO1 family (GmHO1 and GmHO3). Both genes were mainly expressed in three uninfected cell (UC) clusters (clusters 0, 7, and 11), with particularly elevated expression in cluster 11 (Fig. X1B, C). This spatial expression pattern is consistent with our subcellular localization results. Regarding Lb genes, their nodule-specific expression patterns have been extensively reported in previous studies (Wang et al., 2019, Wang et al., 2022, Liu et al., 2023), and we therefore did not repeat the analysis here.
Fig. X1. Expression profiles of heme oxygenase genes in Medicago truncatula and Glycine max derived from public transcriptome datasets. (A) Relative mRNA level of MtHO1 in different tissues of M. truncatula. Data were retrieved from the database (https://lipm-browsers.toulouse.inra.fr/pub/expressionAtlas/app/mtgeav3/00.reference_dataset/Mtr.5569.1.S1_at). (B) UMAP (Uniform Manifold Approximation and Projection) visualization of 15 identified cell clusters in Glycine max nodules and roots (Liu et al., 2023). (C) Expression patterns of GmHO1 (GLYMA_04G147700) and GmHO3 (GLYMA_06G221900) in the single-nucleus transcriptomes of soybean nodules and roots.
References
A clear discrepancy exists between the weak pLjHO1-LjHO1-GFP signal in lb123 mutant in Figure 2 and the strong upregulation of pLjHO1-GUS expression reported in Figure 1. This contradiction needs to be discussed and clarified by the authors. We appreciate the reviewer’s insightful comment. In WT (MG20), LjHO1 expression gradually increases during nodule maturation (Figure 1A–C), which is consistent with its role in degrading heme released from leghemoglobin breakdown during senescence. In contrast, in the lb123 mutant, nodules undergo premature senescence around 2 weeks after inoculation (Wang et al., 2019). The absence of leghemoglobins leads to a small pool of unbound heme, which in turn induces LjHO1 expression at early stages, explaining the strong upregulation observed by pLjHO1-GUS in young lb123 nodules (Figure 1D–F). However, as senescence progresses, the cellular heme pool becomes depleted, and consequently, LjHO1 expression declines at later stages (Figure 1E, F). Importantly, the nodules shown in Figure 2 were sampled at 4 weeks post-inoculation, when LjHO1 expression is already reduced in lb123. This explains why the GFP signal appears weaker in lb123 compared to MG20 at this stage, despite the earlier GUS staining pattern. Taken together, these data reflect different temporal and regulatory layers of LjHO1 expression at transcriptional and protein levels. References
The reference lack some recent key publications. For example, the study demonstrating that NIN and NIN-like proteins regulate Lb expression (PMID: 34709882). Research on the dynamics of hemoglobins during nodule development, nitrate response, and dark stress in Lotus japonicus (PMID: 37976184). We thank the reviewer for pointing out the omission of recent key references. In response, we have updated the Introduction to include the following content: “Recent studies further revealed that Lb expression is transcriptionally regulated by NIN-like proteins, establishing a direct genetic link between nodule developmental regulators and oxygen-buffering capacity (PMID: 34709882). Moreover, Lb gene expression is highly dynamic, responding not only to nodule developmental stages but also to environmental cues such as nitrate availability and dark stress, thereby highlighting their pivotal role in coordinating symbiotic performance with external conditions (PMID: 37976184).”
The Discussion could be further strengthened. One major function of HO and Lb is regulating oxygen levels to support efficient nitrogen fixation. However, the authors did not discuss how oxygen availability may vary across the different mutants, nor its potential correlation with ARA nitrogenase activity. Furthermore, cell-type-specific CRISPR has been applied in legumes (PMID: 40641645), it would be valuable for the authors to discuss the potential of employing such advanced gene-editing technologies to achieve more precise manipulation of symbiotic nitrogen fixation efficiency in future studies.
We thank the reviewer for raising this important point. Based on previous oxygen microprofiling in LbRNAi lines (Ott et al., 2005), which showed a markedly shallower oxygen gradient and substantially higher free-oxygen levels in the central infected zone when leghemoglobins are depleted, we reasonably infer differential oxygen regimes in our mutants as follows. In ho1 mutants, where Lb is present, loss of LjHO1 is expected to alter heme turnover and may modestly increase the pool of available heme; this could lead to dysregulated heme handling that indirectly perturbs cellular respiration and the net effect on free-oxygen concentration is therefore predicted to be modest and context-dependent. By contrast, in ho1lb123 nodules the absence of Lb is expected to dominate oxygen dynamics, producing a shallow oxygen gradient and higher free-oxygen levels similar to those reported for LbRNAi lines, despite any changes in heme turnover caused by LjHO1 loss. We We plan to perform direct oxygen microprofiling and ATP/ADP measurements in future work to test these predictions. However, we have not directly measured oxygen availability in our mutants, and therefore prefer not to overinterpret in the current Discussion. In addition, following the reviewer’s valuable suggestion, we have expanded the Discussion to highlight the potential of cell-type-specific CRISPR approaches for dissecting LjHO1 function in nodules.
References
Minor comments and edits: Line13-14. This sentence is confusing and should be revised. We thank the reviewer for the helpful suggestion. The sentence has been revised for clarity as follows: Here, we show that Lotus japonicus HO1 (LjHO1) is strongly induced in early-stage Lb-deficient nodules, but its expression gradually decreases during nodule development. Subcellular localization analysis revealed that LjHO1 is plastid-localized in uninfected cells, consistent with its localization in wild-type nodules.
Line 89. reference is missing. We thank the reviewer for pointing this out. Upon careful consideration, we believe that the sentence at line 89 is somewhat redundant. In the revised manuscript, we have removed this sentence.
Line 122. The description of the GUS staining is unclear. It would be helpful to specify the incubation times for each nodule size category and the total number of roots analyzed. We have clarified the GUS staining procedure in the Materials and Methods section. Nodules were vacuum-infiltrated at -12 psi and stained at room temperature in the dark, with incubation time adjusted according to nodule size. Specifically, nodules of different sizes were stained together with their associated roots: large nodules (~1 mm in diameter) were incubated for 5 hours, while medium-sized nodules (~0.6 mm in diameter) were incubated for 3 hours. A total of eight independent root transformations were analyzed for reproducibility.
Figure 2B. The lb2lb3 should be lb123-1. Corrected.
|
Reviewer 2 Report
Comments and Suggestions for Authors
The manuscript, entitled “Dissecting the Functional Interplay Between Heme Oxygenase LjHO1 and Leghemoglobins in Lotus japonicus Nodules,” clearly and comprehensively describes the functional interaction between LjHO1 and leghemoglobins (Lbs) in Lotus japonicus during nodule development and senescence. The experimental approach is robust, combining genetics (single and combinatorial mutants), histochemical analysis, UPLC-MS/MS, confocal microscopy, and functional nitrogen fixation assays.
The work is scientifically relevant because it addresses the regulation of heme homeostasis in nodules, an important aspect of plant biology and symbiosis. The methodology is described in sufficient detail for reproducibility, and the results are well organized. However, I suggest some improvements and clarifications:
Introduction:
I consider it necessary to incorporate a brief agronomic context in the introduction to justify why understanding the regulation of heme and leghemoglobins in nodules is important, not only at a basic level but also for agricultural applications. To this end:
- Mention that Lotus japonicus is cultivated or studied in China primarily as a model plant for legume research and nitrogen-fixing symbiosis.
- Highlight that legume crop improvement has an impact on Chinese agriculture, for example, on soil fertility, reduced nitrogen fertilizer use, and sustainability.
Materials and Methods:
- Lines 87-88: Briefly describe the mutants.
- Lines 116-117: Briefly describe.
Results:
- Figure 3 B and E: Are the error bars too large, which could indicate high variability or low replication? It is advisable to review the statistical methods or increase the number of replicates if possible. How is it possible for there to be a statistically significant difference? Could you show the data?
- Some results mention non-significant differences (e.g., heme in ho1lb123 vs. lb123), but it would be helpful to include confidence intervals or show more replicates to reinforce conclusions.
Discussion of Results
-Line 300: Numerical data and percentages (e.g., 17.1% reduction, ARA, etc.) should be removed from the discussion; these belong in the results section.
-Figures should not be described in the discussion; only interpreted and contextualized. Therefore, remove: (Figure X) from the discussion.
- Finally, conclude by reinforcing the discussion on the importance and contributions of the study: currently, it is not entirely clear what new knowledge is contributed and how this impacts the field of symbiosis biology and heme homeostasis.
The manuscript is of high experimental quality and scientific relevance, and with the indicated minor corrections, it would be suitable for publication. It is essential to improve the clarity of the discussion, strengthen the interpretation of the study's significance, and adjust the statistical presentation to make some of the results more convincing.
Author Response
The manuscript, entitled “Dissecting the Functional Interplay Between Heme Oxygenase LjHO1 and Leghemoglobins in Lotus japonicus Nodules,” clearly and comprehensively describes the functional interaction between LjHO1 and leghemoglobins (Lbs) in Lotus japonicus during nodule development and senescence. The experimental approach is robust, combining genetics (single and combinatorial mutants), histochemical analysis, UPLC-MS/MS, confocal microscopy, and functional nitrogen fixation assays.
The work is scientifically relevant because it addresses the regulation of heme homeostasis in nodules, an important aspect of plant biology and symbiosis. The methodology is described in sufficient detail for reproducibility, and the results are well organized. However, I suggest some improvements and clarifications:
Introduction:
I consider it necessary to incorporate a brief agronomic context in the introduction to justify why understanding the regulation of heme and leghemoglobins in nodules is important, not only at a basic level but also for agricultural applications. To this end:
- Mention that Lotus japonicus is cultivated or studied in China primarily as a model plant for legume research and nitrogen-fixing symbiosis.
- Highlight that legume crop improvement has an impact on Chinese agriculture, for example, on soil fertility, reduced nitrogen fertilizer use, and sustainability.
We thank the reviewer for this constructive suggestion. In the revised manuscript, we have incorporated a brief agronomic context in the introduction to better emphasize the importance of our study beyond basic science (Lines 52–56).
Materials and Methods:
- Lines 87-88: Briefly describe the mutants.
Briefly, ho1-1 carries a loss-of-function mutation in LjHO1 leading to reduced heme degradation, whereas lb123-1 is a triple knockout mutant lacking Lb1, Lb2, and Lb3, resulting in severely impaired leghemoglobin accumulation and nitrogen fixation capacity.
- Lines 116-117: Briefly describe.
The 2.9-kb promoter region of LjHO1 was cloned into DX2181G-mCherry vectors harboring the GUS gene, and the resulting construct was introduced into L. japonicus roots via Agrobacterium-mediated hairy root transformation for GUS staining analysis.
Results:
- Figure 3 B and E: Are the error bars too large, which could indicate high variability or low replication? It is advisable to review the statistical methods or increase the number of replicates if possible. How is it possible for there to be a statistically significant difference? Could you show the data?
We thank the reviewer for this comment. The error bars in Figure 3B and E represent the standard deviation from independent biological replicates. The sample sizes are sufficient to support statistical analysis (FigX2A, B). Statistical significance was determined using one-way ANOVA followed by multiple comparison tests (p < 0.05) (Fig X2C, D).
Figure 3B (nodule number per plant): MG20 (n = 32), ho1-1 (n = 32), lb123-1 (n = 55), ho123-1 (n = 50)
Figure 3E (shoot fresh weight, g/plant): MG20 (n = 32), ho1-1 (n = 32), lb123-1 (n = 55), ho123-1 (n = 50)

Fig X2. Nodule number and shoot fresh weight analysis
(A–B) Raw data of nodule number and shoot fresh weight are shown; (C–D) the data in (a, b) were analyzed by one-way ANOVA followed by multiple comparison tests using Prism 10.
- Some results mention non-significant differences (e.g., heme in ho1lb123 vs. lb123), but it would be helpful to include confidence intervals or show more replicates to reinforce conclusions.
We thank the reviewer for the comment. Although the difference in heme levels between ho1lb123 and lb123 is not statistically significant, the current number of replicates (n = 3) supports the conclusions. We have clarified in the text that this difference is non-significant and should be interpreted with caution.
Discussion of Results
-Line 300: Numerical data and percentages (e.g., 17.1% reduction, ARA, etc.) should be removed from the discussion; these belong in the results section.
Agree
-Figures should not be described in the discussion; only interpreted and contextualized. Therefore, remove: (Figure X) from the discussion.
We thank the reviewer for the comment. We agree that in the Discussion, the focus should be on interpreting and contextualizing the results rather than repeating descriptive details. In the revised manuscript, we have carefully revised the Discussion to ensure that any references to figures are only used to support interpretations and arguments, rather than to restate the data.
- Finally, conclude by reinforcing the discussion on the importance and contributions of the study: currently, it is not entirely clear what new knowledge is contributed and how this impacts the field of symbiosis biology and heme homeostasis.
We thank the reviewer for this insightful comment. In the revised Discussion and Conclusion, we now emphasize the conceptual advances of our study. Our work uncovers a previously unrecognized regulatory interplay between leghemoglobins (Lbs) and HO1 in nodule heme homeostasis. Specifically, Lb deficiency promotes LjHO1 expression, likely because total heme levels decrease while free heme increases, thereby inducing its transcription. Conversely, loss of LjHO1 leads to a tendency for heme accumulation in the ho1lb123 background, highlighting its key role in maintaining heme turnover and homeostasis. Furthermore, our results indicate that in wild-type nodules, the majority of heme, mostly derived from Lbs released by infected cells, is transported to uninfected cells for degradation during natural senescence. Subcellular localization analysis shows that LjHO1 remains confined to uninfected cells even under Lb-deficient conditions, demonstrating that Lb deficiency does not alter the intercellular heme degradation pathway. Taken together, these findings reveal a dynamic, developmentally regulated balance between heme synthesis and degradation that is essential for sustaining effective symbiotic nitrogen fixation. By clarifying how legumes actively coordinate heme turnover, our study advances understanding of heme metabolism in nodules and provides broader insights into the regulation of cofactors critical for symbiosis.
The manuscript is of high experimental quality and scientific relevance, and with the indicated minor corrections, it would be suitable for publication. It is essential to improve the clarity of the discussion, strengthen the interpretation of the study's significance, and adjust the statistical presentation to make some of the results more convincing.
We thank the reviewer for the constructive comments and have revised the discussion, strengthened the interpretation, and updated the statistical presentation accordingly.